# Heart rate recovery in 1 minute after the 6-minute walk test predicts adverse outcomes in pulmonary arterial hypertension

**Camila Farnese Rezende**[1], **Eliane Viana Mancuzo**[1,2], **Ricardo de Amorim Corrêa**[1,2] *

1 Postgraduate in Sciences Applied to Adult Health Care, School of Medicine, Hospital das Clínicas, Universidade Federal de Minas Gerais, Belo Horizonte, Minas Gerais, Brazil, 2 Department of Internal Medicine, Pulmonary Section, School of Medicine, Hospital das Clínicas, Universidade Federal de Minas Gerais, Belo Horizonte, Minas Gerais, Brazil

☯ These authors contributed equally to this work.

* racorrea9@gmail.com

**Data Availability Statement:** All relevant data are within the manuscript and its figures.

**Funding:** The authors received no specific funding for this work.

## Abstract

Heart rate recovery in 1 minute (HRR1) after the end of the 6-minute walk test (6MWT) is a non-invasive method of determining autonomic dysfunction. This parameter remains largely unexplored in pulmonary arterial hypertension (PAH) registries. We aimed to define the cut-off value and accuracy for abnormal HRR1 after the 6MWT and to investigate the association between HRR1 and clinical worsening in patients with PAH. This composite outcome was defined as first occurrence of all-cause death OR hospitalization from any cause OR disease progression characterized by decreased ≥ 15% in six-minute walking distance from baseline AND start of new specific PAH treatment or persistent worsening of World Health Organization functional class (WHO-FC). We performed a prospective cohort study that included 102 consecutive patients with PAH confirmed by right heart catheterization that underwent an 6MWT upon the diagnosis, recruited from September 2004 to April 2020 and followed up until April 2021 or death. The median HRR1 was 18 beats (IQR: 10–22), 50 and 52 PAH patients with <18 beats and ≥18 beats, respectively. The best cut-off for HRR1 to discriminate clinical worsening was 17 beats, with area under the curve (AUC) of 0.704 (95%CI: 0.584–0.824). The internal validation model by bootstrap showed an AUC of 0.676 (95%CI: 0.566–0.786) and the most accurate value was obtained in the seventh year of follow-up (AUC = 0.711; 95%CI: 0.596–0.844). Patients with an HRR1 <18 beats at baseline had a median event-free time of 2.17 years (95%CI: 1.82 to 2.52) versus 4.75 years (95% CI: 1.43 to 8.07) from those with ≥18 beats. In conclusion, a HRR1 value of less than 18 beats may be a reliable indicator of poor prognosis in patients with PAH.

## Introduction

Pulmonary arterial hypertension (PAH) is a rare and devastating condition [1]. Risk scores have been developed to predict the prognosis and guide PAH-targeted therapy and are calculated based on clinical features, exercise, echocardiography and hemodynamics [1–4].

**Competing interests:** The authors have declared that no competing interests exist regarding this issue.

The 6-minute walk test (6MWT) has been widely used in exercise capacity evaluation. Walked distance (6MWD) is a key variable in the assessment of PAH prognosis [5] since it correlates with the cardiopulmonary exercise test (CPET) at the time of diagnosis [6]. Heart rate recovery in 1 min (HRR1) after the end of the 6MWT is a candidate tool for adding value in the prediction of PAH prognosis [7–11], as it points to ongoing autonomic dysfunction in conditions such as congestive heart failure, chronic obstructive pulmonary disease and idiopathic pulmonary fibrosis [5, 12, 13]. Although a reduced HRR1 has been associated with morbidity [7, 8] and mortality [9–11] in these conditions, the cut-off value that better discriminates prognosis has not been established but appears to be between ≤16beats [7–9] and ≤18beats [10, 11].

In this pilot study, we aimed to verify a cut-off value of HRR1 that could be associated with adverse outcomes and to assess factors associated with reduced HRR1 in a selected sample of patients with PAH.

## Materials and methods

### Study design

It is a prospective and observational cohort of participants with PAH recruited from the Pulmonary Circulation Unit of *Hospital das Clínicas* of *the Federal University of Minas Gerais* in Belo Horizonte, Brazil, from September 2004 to April 2020 and were followed up until April 2021 or death. Clinical and laboratory assessments, N-terminal pro-brain-type natriuretic peptide (NT-proBNP), echocardiography, right heart catheterization (RHC) variables and the Prospective Registry of Newly Initiated Therapies for Pulmonary Hypertension (COMPERA) risk score were recorded before the initiation of PAH therapy [1, 2, 4]. Participants were followed up every 3 to 6 months throughout the study and updates on medical status were documented per standardised form.

### Study population

For inclusion patients were required to have ≥18 years old and newly hemodynamically confirmed diagnosis of idiopathic PAH (IPAH) or classified into PAH subgroups (schistosomiasis —SchPAH, congenital heart disease—CHDPAH, connective tissue disease—CTDPAH, portopulmonary hypertension—PoPH and HIV infection—HIVPAH) and who were able to perform the 6MWT. Exclusion criteria were patients using β-blockers or PAH-specific therapy before 6MWT and RHC at the baseline test, other causes of PH, pregnancy and if they were prevalent cases. This study was approved by the UFMG's Research Ethics Committee (ETIC nr. 1.057.219/2015) according to the Declaration of Helsinki. All information obtained was considered confidential and the reports and results of this study are presented without any form of individual identification. All participants provided informed consent.

### Six–minute walk test

The 6MWT tests were performed upon confirmed PAH diagnosis, just before beginning specific treatment, and was in accordance with international standards, using a corridor of 30 meters [5]. We recorded saturation by pulse oximetry (SpO$_2$), heart rate (HR), respiratory rate (RR), Borg dyspnoea score before and at the end of the test, HRR1, 6MWD (absolute and percentage of the predicted value calculated by the reference equation in the Brazilian population) [5, 14]. Supplementary oxygen was used when indicated and values of desaturation ≥ 4% was considered significant [5].

## Outcomes

The outcome was time to the first morbimortality event that was composed of all-cause death, hospitalization for any cause or disease progression (decrease of ≥15% in 6MWD from baseline AND worsening of World Health Organization functional class (WHO-FC) or need for additional PAH therapy).

## Statistical analysis

A statistical software was used for data analysis (SPSS, version 23.0, Armonk, NY: IBM Corp). Results are presented as frequency and proportions, mean (SD) or median (IQ range), as indicated. Independent *t* test or Mann-Whitney test and Pearson chi-squared test or Fisher exact test, as appropriate, were used to assess the association between HRR1 (binary) and continuous or categorical variables, respectively. The HRR1 admission value with the best accuracy in predicting the outcome was selected by the receiver operator characteristic (ROC) curve. We examined the internal validation of the model using bootstrap with replacement sampling with 1000 bootstrap samples [15]. A p-value of <0.05 was considered significant. Kaplan-Meier curves and time-dependent ROC curves were used to assess the prognostic value of HRR1 at admission for PAH patients, and statistical significance was tested using the log-rank method.

## Results

### Baseline characteristics at time of diagnosis

A total of 102 consecutive patients out of 109 were followed up for a median of 2.42 years (IQR: 1.08–5.29). The mean age was 48 (SD = 15) years, 68.6% were women and were in FC II (41.2%) and III (44.1%). SchPAH (29.4%), IPAH (23.5%), CHDPAH (20.6%), CTDPAH (15.7%), PoPH (6.9%) and HIVPAH (3.9%) were the prevailing aetiologies. The median baseline HRR1 was 18 beats (IQR: 10–22), 50 patients had <18 beats and 52 patients had ≥18 beats and these findings were similar among the subgroups (p = 0.640) (Table 1). Initial PAH treatment after RHC was monotherapy in 97% of cases that were modified sequentially to combined therapy in 61% of patients in the follow-up.

Patients with HRR1 ≥18beats had better WHO-FC, PAH risk stratification, % predicted diffusion capacity of carbon monoxide (%DLCO) and 6MWD, higher frequency of oxygen use and of 6MWT interruption when compared to HRR1 <18 beats (Table 1). There was no significant between-group difference in the echocardiography and hemodynamics parameters.

### Accuracy of HRR1 in predicting clinical worsening

The ROC curve presented a regular discriminatory power, with an area under the curve (AUC) of 0.704 (95%CI: 0.584–0.824) for HRR1 <18 beats. HRR1 with the best discriminatory power for the prediction of the outcome was 17 beats (sensitivity = 0.83, specificity = 0.56, negative predictive value [NPV] = 1.00, positive predictive value [PPV] = 0.009) (Fig 1A). The internal validation model by bootstrap showed an AUC of 0.676 (95%CI: 0.566–0.786) and the cut-off value of 17 beats had a sensitivity of 0.66, specificity of 0.78, NPV of 1.00 and PPV of 0.002 (Fig 1B). The maximum accuracy value for this combined outcome was obtained from the 7-year onwards follow-up (AUC = 0.711; 95%CI: 0.596–0.844) (Fig 1C).

### Survival analysis

Patients with an HRR1 <18 beats at baseline had increased risk of all endpoint composites compared to ≥18 beats, independent of other 6MWT variables (median event-free time: 2.17

**Table 1. Baseline characteristics of pulmonary arterial hypertension patients with heart rate recovery in 1 minute after the end of the 6-minute walk test < or ≥ 18 beats at diagnosis.**

| Variables at diagnosis | Overall | HRR1 <18beats | HRR1 ≥18beats | p value |
|---|---|---|---|---|
| | (n = 102) | (n = 50) | (n = 52) | |
| Age, yr—mean (SD) | 48 (15) | 48 (16) | 43 (14) | 0.120 |
| Female sex—n (%) | 70 (68.6%) | 33 (66%) | 37 (71.2%) | 0.377 |
| BMI, Kg/m$^2$—median (IQR) | 25 (22–29.3) | 25 (22–30) | 25 (22–29) | 0.240 |
| **PAH subgroup—n (%)** | | | | 0.640 |
| SchPAH | 30 (29.4%) | 14 (28%) | 16 (30.8%) | |
| IPAH | 24 (23.5%) | 11 (22%) | 13 (25.1%) | |
| CHDPAH | 21 (20.6%) | 11 (22%) | 10 (19.2%) | |
| CTDPAH | 16 (15.7%) | 9 (18%) | 7 (13.5%) | |
| PoPH | 7 (6.8%) | 4 (8%) | 3 (5.7%) | |
| HIVPAH | 4 (4%) | 1 (2%) | 3 (5.7%) | |
| **Comorbidities—n (%)** | | | | 0.120 |
| • Ischemic heart disease | 6 (5.8%) | 3 (6%) | 3 (5.7%) | |
| • Hypertension | 22 (21.6%) | 15 (30%) | 7 (13.5%) | |
| • Obesity | 21 (20.6%) | 14 (28%) | 7 (13.5%) | |
| • Atrial fibrillation | 8 (7.8%) | 5 (10%) | 3 (5.7%) | |
| • Diabetes | 8 (7.8%) | 4 (8%) | 4 (7.7%) | |
| • Hypothyroidism | 11 (10.8%) | 6 (12%) | 5 (9.6%) | |
| Former smokers–n (%) | 21 (20.6%) | 13 (26%) | 8 (15.3%) | 0.626 |
| **WHO FC—n (%)** | | | | 0.023* |
| I | 7 (6.9%) | 2 (4%) | 5 (9.6%) | |
| II | 42 (41.2%) | 13 (26%) | 29 (55.8%) | |
| III | 45 (44.1%) | 29 (58%) | 16 (30.8%) | |
| IV | 8 (7.8%) | 6 (12%) | 2 (3.8%) | |
| Time from symptoms to diagnosis, yr -median (IQR) | 2 (1–4.3) | 2 (1–4.5) | 2 (1–4.5) | 0.706 |
| **COMPERA risk stratification—n(%)** | | | | 0.001* |
| • Low risk | 32 (31.3%) | 6 (12%) | 26 (50%) | |
| • Intermediate risk | 37 (36.3%) | 20 (40%) | 17 (32.7%) | |
| • High risk | 33 (32.4%) | 24 (48%) | 9 (17.3%) | |
| **Prior medications—n(%)** | | | | |
| • Oxygen | 10 (9.8%) | 10 (20%) | 0 | 0.002* |
| • Diuretic | 40 (39.2%) | 24 (48%) | 16 (30.7%) | 0.320 |
| **First-line target PAH therapy—n (%)** | | | | 0.589 |
| • Monotherapy | 99 (97%) | 48 (96%) | 51 (98%) | |
| • Combination | 1 (1%) | 1 (2%) | 0 | |
| • Calcium-channel blocker | 2 (2%) | 1 (2%) | 1 (2%) | |
| NT-proBNP,ng.L-1—median (IQR) | 597 (280–1566) | 614 (420–1200) | 597 (330–890) | 0.120 |
| • PaO2, mmHg—mean (SD) | 72.5 (13.4) | 71.4 (13.6) | 73.8 (13.1) | 0.240 |
| • PaCO2, mmHg—mean (SD) | 32.5 (4.1) | 32.8 (4.0) | 32.2 (4.4) | 0.235 |
| eGFR by CKD-EPI—mean (SD) | 91.4 (18.7) | 91.6 (18.6) | 91.7 (18.8) | 0.210 |
| **Pulmonary function** | | | | |
| • %DLCO—median (IQR) | 70 (57.7–80) | 60 (55–76) | 72 (64–80) | 0.040* |
| **6MWT** | | | | |
| • 6MWD, m—median (IQR) | 428.9 (316.8–510.7) | 384 (240.4–478.1) | 462 (396.9–551.2) | 0.001* |
| • % predicted 6MWD—median (IQR) | 77 (62.8–87.1) | 75 (55–83) | 82 (70–91) | 0.007* |
| • Desaturation ≥ 4%—n (%) | 80 (78%) | 45 (78.4%) | 35 (67.3%) | 0.190 |

(*Continued*)

**Table 1.** (Continued)

| Variables at diagnosis | Overall | HRR1 <18beats | HRR1 ≥18beats | p value |
|---|---|---|---|---|
| | (n = 102) | (n = 50) | (n = 52) | |
| • Baseline HR, beats—median (IQR) | 82 (72–97) | 87 (76–95) | 81 (74–91) | 0.206 |
| • Peak HR, beats—median (IQR) | 127 (110–148) | 125 (109–140) | 129 (114–141) | 0.329 |
| • HRR1, beats—median (IQR) | 18 (10–22) | 11 (9–15) | 23 (20–30) | <0.001* |
| • Stopped during the test—n (%) | 13 (12.7%) | 10 (20%) | 3 (5.7%) | 0.038* |
| **Transthoracic echocardiogram** | | | | |
| sPAP, mmHg—median (IQR) | 75 (60–94) | 74 (62–91) | 82 (59–96) | 0.587 |
| RAP, mmHg—median (IQR) | 10 (10–15) | 10 (10–15) | 10 (5–13.5) | 0.142 |
| TRV, m/s—mean (SD) | 4.04 (0.70) | 3.99 (0.68) | 4.11 (0.73) | 0.374 |
| TAPSE, mm—median (IQR) | 14 (12.8–19.0) | 14 (12–19) | 16 (13–19) | 0.468 |
| Pericardial effusion—n(%) | 14 (13.9%) | 10 (20%) | 4 (8%) | 0.166 |
| **Right heart catheterization** | | | | |
| sPAP, mmHg—median (IQR) | 86 (70–105) | 87 (71–102) | 82 (70–107.5) | 0.733 |
| mPAP, mmHg—median (IQR) | 51 (42–62.5) | 50 (42–64.5) | 52 (41.5–66) | 0.973 |
| RAP, mmHg—median (IQR) | 8.5 (5–12) | 9 (5–12) | 8 (5–10) | 0.361 |
| PCWP, mmHg—median (IQR) | 10 (7.8–12) | 10 (7–12) | 10 (8–12.5) | 0.640 |
| CI, L.min$^{-1}$.m$^{-2}$—median (IQR) | 2.4(1.8–3.1) | 2.2 (1.9–3.0) | 2.7 (1.8–3.4) | 0.167 |
| PVR, dynes/sec/cm$^{-5}$- median (IQR) | 980 (630–1394) | 1040 (622–1410) | 920 (592–1296) | 0.405 |
| SvO2, %—mean (SD) | 64.9 (7.4) | 63.8% (7.1%) | 69% (7.6%) | 0.060 |

Data are expressed as the mean (SD: standard deviation) or median (IQR: interquartile range). 6MWD: six-minute walk distance; 6MWT: six-minute walk test; BMI: body mass index; CHDPAH: congenital heart disease associated with pulmonary arterial hypertension; CI: cardiac index; COMPERA: Comparative, Prospective Registry of Newly Initiated Therapies for Pulmonary Hypertension; CTDPAH: connective tissue disease associated with pulmonary arterial hypertension; eGFR by CKD-EPI: estimated glomerular filtration rate by Chronic Kidney Disease Epidemiology Collaboration; %DLCO: % predicted diffusion capacity of carbon monoxide; HIVPAH: human immunodeficiency virus associated with pulmonary arterial hypertension; HR: heart rate; HRR1: heart rate recovery at 1 minute; IPAH: idiopathic pulmonary arterial hypertension; mPAP: mean pulmonary arterial pressure; NT-proBNP: N-terminal pro-brain-type natriuretic peptide; PAH: pulmonary arterial hypertension; PaCO2: arterial carbon dioxide pressure; PaO2: arterial oxygen pressure; PCWP: pulmonary capillary wedge pressure; PoPH: portopulmonary hypertension; PVR: pulmonary vascular resistance; RAP: right atrial pressure; SchPAH: schistosomiasis associated with pulmonary arterial hypertension; sPAP: systolic pulmonary arterial pressure; SvO2: mixed venous oxygen saturation; TAPSE: tricuspid annular plane systolic excursion; TRV: tricuspid regurgitation velocity; WHO FC: World Health Organization functional class.

*p<0.05 for the comparison between PAH patients with HHR1 <18beats and ≥18beats.

years, 95%CI: 1.82 to 2.52 vs 4.75 years, 95%CI: 1.43 to 8.07; log-rank: p<0.001, respectively) (Fig 2). The distribution of outcomes was disease progression (55%), hospitalization (20%), death (1%) and 24% had no event during the follow-up time. The main causes of death were right ventricular failure (n = 25) followed by sepsis (n = 7), liver failure (n = 4), pulmonary embolism (n = 1) and pulmonary artery dissection (n = 1).

## Discussion

In the present cohort, an HRR1 of <18 beats at baseline had a high negative predictive value for predicting all-cause mortality, hospitalization and disease progression in the participants.

HRR1 reflects the progressive increase in the vagal tone parallel to the decline of sympathetic stimulation. Low HRR1 is a non-invasive marker of autonomic dysfunction [12, 13] with an HRR1 of <18 beats being associated with a worse prognosis. The reduced cardiac output due right ventricle (RV) dysfunction in patients with more severe disease with consequent sympathetic hyperactivity and reduced systemic and local parasympathetic activity were possible mechanisms involved in this response [16, 17]. Autonomic responses to exercise, such as

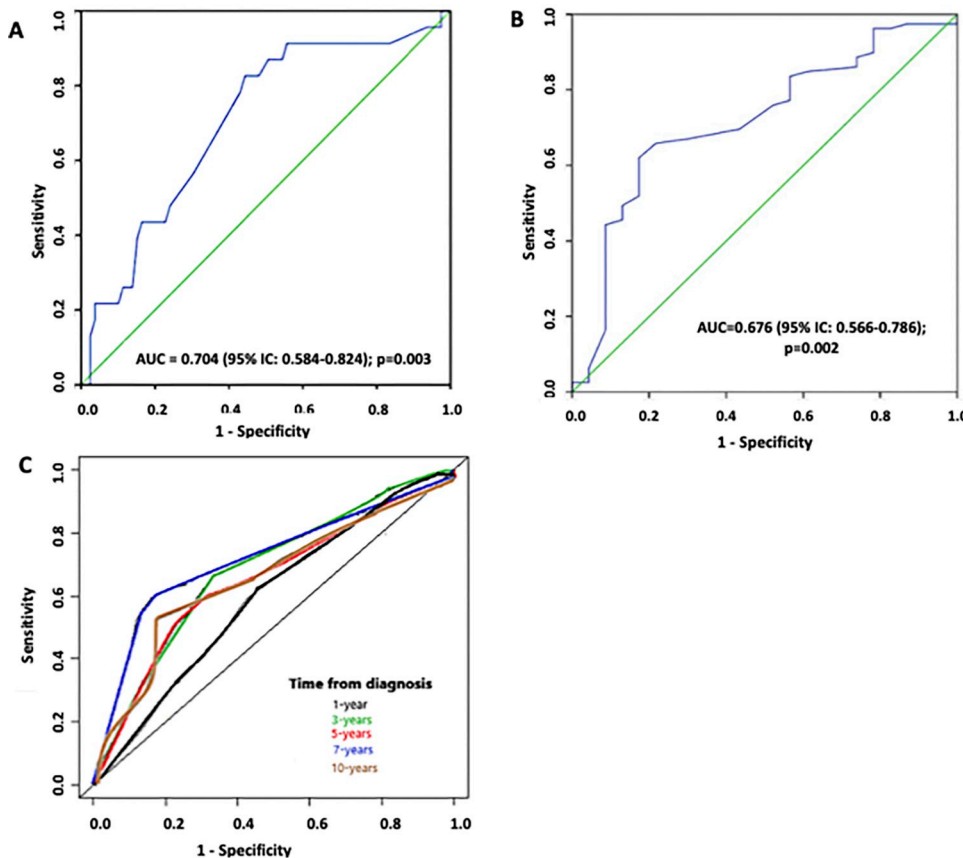

**Fig 1. Prognosis accuracy of the HRR1 admission value (cut-off point = 17 beats) for predicting clinical worsening in PAH.** (A) Receiver operator characteristic (ROC) curve of heart rate recovery in 1 minute (HRR1) after the end of six-minute walk test in all pulmonary arterial hypertension (PAH) patients enrolled in the study. (B) ROC curve representing the internal validation model by bootstrap. (C) Time-dependent ROC curves representing the prognostic accuracy of the HRR1 admission for predicting of the composite outcome at 1-, 3-, 5-, 7- and 10-years.

chronotropic response, provide prognostic information in PAH [7–11, 18]. However, the exact HRR1 that predicts morbidity and mortality events has not yet been definitively established. Table 2 provides the studies that associated HHR1 with clinical outcome in PAH. An HRR1 of <16 beats in the 6MWT was a stronger predictor of clinical worsening in patients with IPAH and CTDPAH than the 6MWD [7, 8]. Previous studies have reported that a HRR1 of ≤18 beats in 72 and 418 PAH patients in the CPET and incremental shuttle walking test, respectively, was the only independent predictor of mortality (log-rank test, p<0.05) [10, 11]. In the present study, the baseline HRR1 of <18 beats in the 6MWT was closer to that of other exercise tests [6, 10, 11, 17].

Reduced HRR1 was associated with worse WHO-FC and gas exchange, oxygen use, intermediate or high PAH risk stratification and decreased exercise capacity. These associations have been previously reported using CPET, a gold-standard method that assess the true cardiopulmonary functional status of patients [6, 10, 18–20], but it is available only at a few centers. The present finding reinforces HRR1 as an independent biomarker of PAH severity and is an accessible, simple to perform, reliable and safe variable obtained through the 6MWT [5]. Despite the hypothesis that the mechanism of reduced HRR1 in PAH is associated with RV dysfunction [17], there was no significant difference between-groups in tricuspid annular

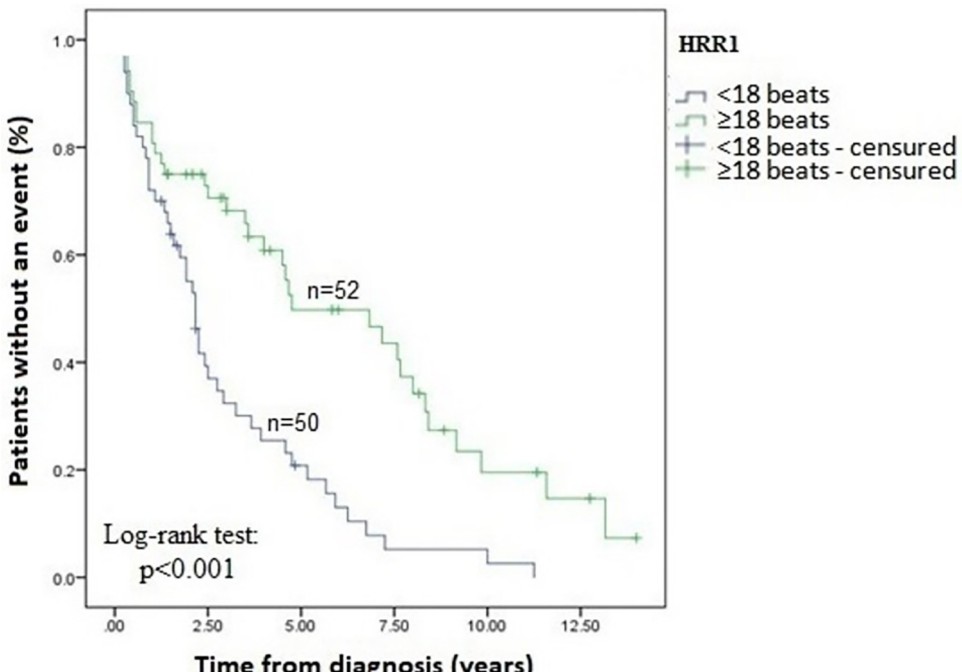

**Fig 2. Event-free survival estimate of the pulmonary arterial hypertension cohort based on the HRR1 <18 beats (n = 50) and ≥18 beats (n = 52) after a long follow-up period.** HRR1: heart rate recovery in 1 minute.

**Table 2. Studies that investigated the association between HRR1 and clinical outcome in pulmonary arterial hypertension patients.**

| Characteristic | Minai OA (2012) [7] | Minai OA (2015) [8] | Ramos RP (2012) [10] | Billings CG (2017) [11] | Our study |
|---|---|---|---|---|---|
| Study design and time period | Retrospective 2009–2010 | Retrospective 2009–2011 | Retrospective Follow up by a median of 28 months | Retrospective 2001–2010 | Prospective 2004–2021 |
| PAH subgroups–n (%) | | | | | |
| • IPAH | 75 (100%) | | 72 (100%) | 418 (100%) | 102 (100%) |
| • CTDPAH | | 66 (100%) | 37 (51.4%) | 133 (31.8%) | 24 (23.5%) |
| • CHDPAH | | | 18 (25%) | 144 (34.5%) | 16 (15.7%) |
| • SchPAH | | | 5 (6.9%) | 119 (28.4%) | 21 (20.6%) |
| • PoPH | | | 4 (5.6%) | | 30 (29.4%) |
| • HIVPAH | | | 5 (6.9%) | | 7 (6.8%) |
| • Other | | | 3 (4.2%) | 22 (5.3%) | 4 (4%) |
| Patients with PAH-specific therapy at the time of exercise test–n (%) | 71 (94.6%) | 50 (75.8%) | 26 (36%) | 0 | 0 |
| Exercise test | 6MWT | 6MWT | CPET | ISWT | 6MWT |
| HRR1 cut-off point | < 16 beats | < 16 beats | ≤ 18 beats | ≤ 18 beats | < 18 beats |
| Outcome | Clinical worsening | Clinical worsening | Mortality | Mortality | Clinical worsenig |
| Survival analysis (hazard ratio or median survival time / p value) | HR: 5.2 (95% CI: 1.8–14.8) p = 0.002 | HR: 6.4 (95% CI: 2.6–19.2) p<0.0001 | HR: 1.19 (95% CI: 1.03–1.37) p<0.05 | p = 0.04 | Median survival time: 2.17y (<18 beats) versus 4.75y (≥18beats) p<0.001 |

6MWT: six-minute walk test; CHDPAH: congenital heart disease associated with pulmonary arterial hypertension; CPET: cardiopulmonary exercise test; CTDPAH: connective tissue disease associated with pulmonary arterial hypertension; HIVPAH: human immunodeficiency virus associated with pulmonary arterial hypertension; HRR1: heart rate recovery at 1 minute; IPAH: idiopathic pulmonary arterial hypertension; ISWT: incremental shuttle walk test; PAH: pulmonary arterial hypertension; PoPH: portopulmonary hypertension; SchPAH: schistosomiasis associated with pulmonary arterial hypertension.

plane systolic excursion and cardiac index measures. This might be explained by the fact that echocardiography and RHC are performed at rest and this could be changed if the measurements were performed under exercise [21].

The study is limited by its single-center design and having SchPAH as our most prevalent aetiology impairs its further generalization, although SchPAH has been considered one of the most prevalent PAH etiology worldwide [1, 22]. It is worth mentioning that 6MWT and RHC examinations were performed at an average period of 3-month intervals, which reflects the context of usual clinical practice. However, this is the first cohort study addressing this topic with a longer follow-up time, and the specific treatment for PAH was started only after diagnosis by RHC.

In conclusion, a delay in HRR1 of <18 beats predicts risk of the occurrence of clinical outcomes in PAH, irrespective of the initial 6MWD. The usefulness of this variable in stratification of risk in PAH patients needs to be further investigated in others prospective studies.

## Supporting information

**S1 Checklist. STROBE statement—checklist of items that should be included in reports of** *cohort studies.*
(DOC)

## Author Contributions

**Conceptualization:** Camila Farnese Rezende, Eliane Viana Mancuzo, Ricardo de Amorim Corrêa.

**Data curation:** Camila Farnese Rezende, Eliane Viana Mancuzo, Ricardo de Amorim Corrêa.

**Formal analysis:** Camila Farnese Rezende, Eliane Viana Mancuzo, Ricardo de Amorim Corrêa.

**Investigation:** Camila Farnese Rezende, Eliane Viana Mancuzo, Ricardo de Amorim Corrêa.

**Methodology:** Camila Farnese Rezende, Eliane Viana Mancuzo, Ricardo de Amorim Corrêa.

**Validation:** Camila Farnese Rezende, Eliane Viana Mancuzo, Ricardo de Amorim Corrêa.

**Writing – original draft:** Camila Farnese Rezende, Eliane Viana Mancuzo, Ricardo de Amorim Corrêa.

**Writing – review & editing:** Camila Farnese Rezende, Eliane Viana Mancuzo, Ricardo de Amorim Corrêa.

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
