## [Decision Letter · Decision Letter 0]

4 Mar 2022

PONE-D-21-37582Heart rate recovery in 1 minute after the 6-minute walk test predicts adverse outcomes in pulmonary arterial hypertensionPLOS ONE

Dear Dr. Corrêa,

Thank you for submitting your manuscript to PLOS ONE. After careful consideration, we feel that it has merit but does not fully meet PLOS ONE’s publication criteria as it currently stands. Therefore, we invite you to submit a revised version of the manuscript that addresses the points raised during the review process.

We look forward to receiving your revised manuscript.

Kind regards,

Jeffrey S Isenberg, MD, MPH

Academic Editor

PLOS ONE

“The authors have declared that no competing interests exist regarding this issue.”

Additional Editor Comments:

The expert Reviewers were positively disposed to the study but have specific suggestions that require addressing and this may require additional data/information.

Reviewers' comments:

Reviewer's Responses to Questions

**Comments to the Author**

1. Is the manuscript technically sound, and do the data support the conclusions?

Reviewer #1: Yes

Reviewer #2: Yes

2. Has the statistical analysis been performed appropriately and rigorously? 

Reviewer #1: Yes

Reviewer #2: Yes

3. Have the authors made all data underlying the findings in their manuscript fully available?

Reviewer #1: Yes

Reviewer #2: Yes

4. Is the manuscript presented in an intelligible fashion and written in standard English?

Reviewer #1: Yes

Reviewer #2: Yes

5. Review Comments to the Author

Reviewer #1: The authors present a brief, interesting and useful analysis of individuals with recently dx PAH comparing heart rate 1 minute after completing a six-minute walk and clinical changes. The study was prospective and spanned over 15 years with 102 subjects. Lower hear rate was linked to worse clinical outcomes. A bootstrap method was used to validate the findings. The paper is well written and supported by a table and figures.

The Methods section needs a description of inclusion/exclusion criteria, the study institutional approval number/information, and details on obtaining informed consent and protected management of human subject data.

The authors might consider increasing the data on the cohort. Such details as BMI, associated medical conditions, tobacco usage, cause of death, etc. would be of interest.

The authors might put together a second table that summarizes other studies of similar design and intent. The reader would find this a helpful addition to the author's original work and could be mentioned in the Discussion section.

Table 1 may have a few abbreviations not spelled out in the legend. Please correct.

The figures are quite blurry and should be sharped up and improved.

Reviewer #2: The paper is very interesting and important for the area. I suggest that the authors complete Table 1 with the medications used, the number of patients who used oxygen, how many stopped during the six-minute walk test, or if others also stopped. Adding how many desaturate during the test, this is very relevant. I suggest adding the percentage of the predicted distance traveled for the population studied.

6. PLOS authors have the option to publish the peer review history of their article (what does this mean?). If published, this will include your full peer review and any attached files.

Reviewer #1: No

Reviewer #2: No

---

## [Author Response · Author response to Decision Letter 0]

18 Apr 2022

REVIEWER 1:

1. The Methods section needs a description of inclusion/exclusion criteria, the study institutional approval number/information, and details on obtaining informed consent and protected management of human subject data.

A - We agreed and did the changeas suggested.

2. The authors might consider increasing the data on the cohort. Such details as BMI, associated medical conditions, tobacco usage, cause of death, etc. would be of interest.

A - We agreed and added BMI, comorbidities, previous tobacco usage as suggested in table 1. 

 In page 9, line 182, we added the causes of death. 

3. The authors might put together a second table that summarizes other studies of similar design and intent. The reader would find this a helpful addition to the author's original work and could be mentioned in the Discussion section.

A - We agreed and included table 2 in the discussion section. 

4. Table 1 may have a few abbreviations not spelled out in the legend. Please correct.

A - We corrected the legend. 

5. The figures are quite blurry and should be sharped up and improved.

A – We improved the figures and uploaded the files to the Preflight Analysis and Conversion Engine (PACE) digital diagnostic tool. 

REVIEWER 2 :

1. I suggest that the authors complete Table 1 with the medications used, the number of patients who used oxygen, how many stopped during the six-minute walk test, or if others also stopped. Adding how many desaturate during the test, this is very relevant. I suggest adding the percentage of the predicted distance traveled for the population studied.

A - We agreed and added medications used at the time of the six-minute wak test (diuretic and oxygen), how many stopped and desaturate during this test and the percentage of the predicted distance calculated by the reference equation in the Brazilian population as suggested in table 1. For this reason, we added the reference 14. 

Best,

The authors

---

## [Decision Letter · Decision Letter 1]

10 May 2022

Heart rate recovery in 1 minute after the 6-minute walk test predicts adverse outcomes in pulmonary arterial hypertension

PONE-D-21-37582R1

Dear Dr. Corrêa,

We’re pleased to inform you that your manuscript has been judged scientifically suitable for publication and will be formally accepted for publication once it meets all outstanding technical requirements.

Kind regards,

Nejka Potocnik

Academic Editor

PLOS ONE

Additional Editor Comments (optional): The authors present a brief, interesting and useful analysis of individuals with PAH comparing heart rate 1 minute after completing a six-minute walk and clinical changes. The study was prospective and spanned over 15 years with 102 subjects. Lower hear rate was linked to worse clinical outcomes. A bootstrap method was used to validate the findings. The paper is well written and ready to be bublished.

Reviewers' comments:

Reviewer's Responses to Questions

**Comments to the Author**

1. If the authors have adequately addressed your comments raised in a previous round of review and you feel that this manuscript is now acceptable for publication, you may indicate that here to bypass the “Comments to the Author” section, enter your conflict of interest statement in the “Confidential to Editor” section, and submit your "Accept" recommendation.

Reviewer #1: All comments have been addressed

Reviewer #2: All comments have been addressed

2. Is the manuscript technically sound, and do the data support the conclusions?

Reviewer #1: Yes

Reviewer #2: Yes

3. Has the statistical analysis been performed appropriately and rigorously? 

Reviewer #1: Yes

Reviewer #2: Yes

4. Have the authors made all data underlying the findings in their manuscript fully available?

Reviewer #1: Yes

Reviewer #2: Yes

5. Is the manuscript presented in an intelligible fashion and written in standard English?

Reviewer #1: Yes

Reviewer #2: Yes

6. Review Comments to the Author

Reviewer #1: The authors submitted a revised manuscript. Thye addressed the comments of the Reviewer and are thanked for the effort.

Reviewer #2: (No Response)

7. PLOS authors have the option to publish the peer review history of their article (what does this mean?). If published, this will include your full peer review and any attached files.

Reviewer #1: No

Reviewer #2: No

---

## [Editor Report · Acceptance letter]

16 May 2022

PONE-D-21-37582R1 

Heart rate recovery in 1 minute after the 6-minute walk test predicts adverse outcomes in pulmonary arterial hypertension 

Dear Dr. Corrêa:

I'm pleased to inform you that your manuscript has been deemed suitable for publication in PLOS ONE. Congratulations! Your manuscript is now with our production department. 

Kind regards, 

on behalf of

Dr. Nejka Potocnik 

Academic Editor

PLOS ONE